# Justification of research using systematic reviews continues to be inconsistent in clinical health science—A systematic review and meta-analysis of meta-research studies

**Jane Andreasen**[1]*, **Birgitte Nørgaard**[2], **Eva Draborg**[2], **Carsten Bogh Juhl**[3], **Jennifer Yost**[4], **Klara Brunnhuber**[5], **Karen A. Robinson**[6], **Hans Lund**[7]

1 Department of Physiotherapy and Occupational Therapy, Aalborg University Hospital, Denmark and Public Health and Epidemiology Group, Department of Health, Science and Technology, Aalborg University, Aalborg, Denmark, 2 Department of Public Health, University of Southern Denmark Odense, Denmark, 3 Department of Sports Science and Clinical Biomechanics, University of Southern Denmark and Department of Physiotherapy and Occupational Therapy, Copenhagen University Hospital, Herlev and Gentofte, Herlev, Denmark, 4 M. Louise Fitzpatrick College of Nursing, Villanova University, Villanova, PA, United States of America, 5 Digital Content Services, Elsevier, London, United Kingdom, 6 Johns Hopkins University School of Medicine, Baltimore, MD, United States of America, 7 Department of Evidence-Based Practice, Western Norway University of Applied Sciences, Bergen, Norway

* jaan@rn.dk

## Abstract

### Background

Redundancy is an unethical, unscientific, and costly challenge in clinical health research. There is a high risk of redundancy when existing evidence is not used to justify the research question when a new study is initiated. Therefore, the aim of this study was to synthesize meta-research studies evaluating if and how authors of clinical health research studies use systematic reviews when initiating a new study.

### Methods

Seven electronic bibliographic databases were searched (final search June 2021). Meta-research studies assessing the use of systematic reviews when justifying new clinical health studies were included. Screening and data extraction were performed by two reviewers independently. The primary outcome was defined as the percentage of original studies within the included meta-research studies using systematic reviews of previous studies to justify a new study. Results were synthesized narratively and quantitatively using a random-effects meta-analysis. The protocol has been registered in Open Science Framework (https://osf.io/nw7ch/).

### Results

Twenty-one meta-research studies were included, representing 3,621 original studies or protocols. Nineteen of the 21 studies were included in the meta-analysis. The included studies represented different disciplines and exhibited wide variability both in how the use of previous systematic reviews was assessed, and in how this was reported. The use of

**Data Availability Statement:** All relevant data are within the paper and its Supporting Information files.

**Funding:** The authors received no specific funding for this work.

**Competing interests:** The authors have declared that no competing interests exist.

systematic reviews to justify new studies varied from 16% to 87%. The mean percentage of original studies using systematic reviews to justify their study was 42% (95% CI: 36% to 48%).

## Conclusion

Justification of new studies in clinical health research using systematic reviews is highly variable, and fewer than half of new clinical studies in health science were justified using a systematic review. Research redundancy is a challenge for clinical health researchers, as well as for funders, ethics committees, and journals.

## Introduction

Research redundancy in clinical health research is an unethical, unscientific, and costly challenge that can be minimized by using an evidence-based research approach. First introduced in 2009 and since endorsed and promoted by organizations and researchers worldwide [1–6], evidence-based research is an approach whereby researchers systematically and transparently take into account the existing evidence on a topic before embarking on a new study. The researcher thus strives to enter the project unbiased, or at least aware of the risk of knowledge redundancy bias. The key is an evidence synthesis using formal, explicit, and rigorous methods to bring together the findings of pre-existing research to synthesize the totality what is known [7]. Evidence syntheses provide the basis for an unbiased justification of the proposed research study to ensure that the enrolling of participants, resource allocation, and healthcare systems are supporting only relevant and justified research. Enormous numbers of research studies are conducted, funded, and published globally every year [8]. Thus, if earlier relevant research is not considered in a systematic and transparent way when justifying research, the foundation for a research question is not properly established, thereby increasing the risk of redundant studies being conducted, funded, and published resulting in a waste of resources, such as time and funding [1, 4]. Most importantly, when redundant research is initiated, participants unethically and unnecessarily receive placebos or receive suboptimal treatment.

Previous meta-research, defined as the study of research itself including the methods, reporting, reproducibility, evaluation and incentives of the research [9] have shown that there is considerable variation and bias in the use of evidence syntheses to justify research studies [10–12]. To the best of our knowledge, a systematic review of previous meta-research studies assessing the use of systematic reviews to justify studies in clinical health research has not previously been conducted. Evaluating how evidence-based research is implemented in research practices across disciplines and specialties when justifying new studies will provide an indication of the integration of evidence-based research in research practices [9]. The present systematic review aimed to identify and synthesize results from meta-research studies, regardless of study type, evaluating if and how authors of clinical health research studies use systematic reviews to justify a new study.

## Methods

Prior to commencing the review, we registered the protocol in the Open Science Framework (https://osf.io/nw7ch/). The protocol remained unchanged, but in this paper we have made adjustments to the risk-of-bias assessment, reducing the tool to 10 items and removing the

assessment of reporting quality. The review is presented in accordance with the Preferred Reporting Items for Systematic review and Meta-Analysis (PRISMA) guidelines [13].

## Eligibility criteria

Studies were eligible for inclusion if they were original meta-research studies, regardless of study type, that evaluated if and how authors of clinical health research studies used systematic reviews to justify new clinical health studies. No limitations on language, publication status, or publication year were applied. Only meta-research studies of studies on human subjects in clinical health sciences were eligible for inclusion. The primary outcome was defined as the percentage of original studies within the included meta-research studies using systematic reviews of previous studies to justify a new study. The secondary outcome was how the systematic reviews of previous research were used (e.g., within the text to justify the study) by the original studies.

## Information sources and search strategy

This study is one of six ongoing evidence syntheses (four systematic reviews and two scoping reviews) planned to assess the global state of evidence-based research in clinical health research. These are; a scoping review mapping the area broadly to describe current practice and identify knowledge gaps, a systematic review on the use of prior research in reports of randomized controlled trials specifically, three systematic reviews assessing the use of systematic reviews when justifying, designing [14] or putting results of a new study in context, and finally a scoping review uncovering the breadth and characteristics of the available, empirical evidence on the topic of citation bias. Further, the research group is working with colleagues on a Handbook for Evidence-based Research in health sciences. Due to the common aim across the six evidence syntheses, a broad overall search strategy was designed to identify meta-research studies that assessed whether researchers used earlier similar studies and/or systematic reviews of earlier similar studies to inform the justification and/or design of a new study, whether researchers used systematic reviews to inform the interpretation of new results, and meta-research studies that assessed if there were published redundant studies within a specific area or not.

The first search was performed in June 2015. Databases included MEDLINE via both PubMed and Ovid, EMBASE via Ovid, CINAHL via EBSCO, Web of Science (Science Citation Index Expanded (SCI-EXPANDED), Social Sciences Citation Index (SSCI), Arts & Humanities Citation Index (A&HCI), and the Cochrane Methodology Register (CMR, Methods Studies) from inception (Appendix 1 in S1 File). In addition, reference lists of included studies were screened for relevant articles, as well as the authors' relevant publications and abstracts from the Cochrane Methodology Reviews.

Based upon the experiences from the results of the baseline search in June 2015, an updated and revised search strategy was conducted in MEDLINE and Embase via Ovid from January 2015 to June 2021 (Appendix 1 in S1 File). Once again, the reference lists of new included studies were screened for relevant references, as were abstracts from January 2015 to June 2021 in the Cochrane Methodology Reviews. Experts in the field were contacted to identify any additional published and/or grey literature. No restrictions were made on publication year and language. See Appendix 1 and Appendix 2 in S1 File for the full search strategy.

## Screening and study selection

Following deduplication, the search results were uploaded to Rayyan (https://rayyan.qcri.org/welcome). The search results from the 1st search (June 2015) were independently screened by

a pair of reviewers. Twenty screeners were paired, with each pair including an author very experienced in systematic reviews and a less experienced author. To increase consistency among reviewers, both reviewers initially screened the same 50 publications and discussed the results before beginning screening for this review. Disagreements on study selection were resolved by consensus and discussion with a third reviewer, if needed. The full-text screening was also performed by two reviewers independently. Disagreements on study selection were resolved by consensus and discussion. There were also two independent reviewers who screened following the last search, using the same procedure, as for the first search, for full-text screening and disagreements. The screening procedures resulted in a full list of studies potentially relevant for one or more of the six above-mentioned evidence syntheses.

A second title and abstract screening and full-text screening of the full list was then performed independently by two reviewers using screening criteria specific to this systematic review. Reasons for excluding trials were recorded, and disagreements between the reviewers were resolved through discussion. If consensus was not reached, a third reviewer was involved.

## Data extraction

We developed and pilot tested a data extraction form to extract data regarding study characteristics and outcomes of interest. Two reviewers independently extracted data, with other reviewers available to resolve disagreements. The following study characteristics were extracted from each of the included studies: bibliographic information, study aim, study design, setting, country, inclusion period, area of interest, results, and conclusion. Further, data for this study's primary and secondary outcomes were extracted; these included the percentage of original studies using systematic reviews to justify their study and how the systematic reviews of previous research were used (e.g., within the text to justify the study) by the original studies.

## Risk-of-bias assessment

No standard tool was identified to assess the risk of bias in empirical meta-research studies. The Editorial Group of the Evidence-Based Research Network prepared a risk-of-bias tool for the planned five systematic reviews with list of items important for evaluating the risk of bias in meta-research studies. For each item, one could classify the study under examination as exhibiting a "low risk of bias", "unclear risk of bias" or "high risk of bias". We independently tested the list of items upon a sample of included studies. Following a discussion of the different answers, we adjusted the number and content of the list of items to ten and defined the criteria to evaluate the risk of bias in the included studies (Table 1). Each of the included meta-research studies was appraised independently by two reviewers using the customized checklist to determine the risk of bias. Disagreements regarding the risk of bias were solved through discussion. No study was excluded on the grounds of low quality.

## Data synthesis and interpretation

In addition, to narratively summarizing the characteristics of the included meta-research studies and their risk-of-bias assessments, the percentage of original studies using systematic review of previous similar studies to justify a new study (primary outcome) was calculated as the number of studies using at least one systematic review, divided by the total number of original studies within each of the included meta-research studies. A meta-analysis using the random-effects model (DerSimonian and Laird) was used to estimate the overall estimate and perform the forest plot as this model is the default when using the metaprop command. Heterogeneity was evaluated estimating the $I^2$ statistics (the percentage of variance attributable to

**Table 1. Risk of bias tool.**

| Item | Prompt for high risk of bias |
|---|---|
| 1) Is there a clear and focused aim? | A vague or unclear aim of the study |
| 2) Is there a match between the aim and chosen method(s)? | The method chosen will not or is very unlikely to be able to answer the aim of the meta-research study |
| 3) Was the chosen source the best alternative among others? | No or poor argument for selecting the source and/or no or poor description of why other options were not selected |
| 4) Were all important variables considered? | No or poor argument for selecting the variable(s) and/or no or poor description of why other variable(s) were not selected |
| 5) Were the same variables considered in all sources? | Variables used depended upon the source, and/or the same variables were not extracted from all included sources |
| 6) Was the data collection transparent and data unambiguously identified? | No description or poor description of how data were extracted and/or the data extraction were not performed by two independent reviewers |
| 7) Does the classification of the variables/answers appear unaffected by prior knowledge about the results? | No protocol, and/or registration of the background and methods were prepared and made publicly available |
| 8) Was an appropriate analysis method chosen? | The selected analysis(es) does not match the aim and/or was methodologically not correct/widely accepted and/or relevant for the type of data used in the meta-research study, and/or a widely accepted analysis method was not used without any justification |
| 9) Was any possible systematic error or bias taken into consideration in the data collection and/or analysis? | No discussion of the limitations of the study results were included in the Discussion section, and/or the existing limitations/biases had either no impact upon the conclusion, or there was no explanation of why the limitations/biases did not affect the conclusion |
| 10) Is the conclusion supported by the data? | The conclusion and/or parts of the conclusion includes aspects not supported by the results |

heterogeneity i.e., inconsistency) and the between study variance $tau^2$. When investigating reasons for heterogeneity, a restricted maximum likelihood (REML) model was used and covariates with the ability to reduce $tau^2$ was deemed relevant. [15].

All analyses were conducted in Stata, version 17.0 (StataCorp. 2019. *Stata Statistical Software*: *Release 17*. College Station, TX: StataCorp LLC).

# Results

## Study selection

In total, 30,592 publications were identified through the searches. Of these, 69 publications were determined eligible for one of the six evidence syntheses. A total of 21 meta-research studies fulfilled the inclusion criteria for this systematic review [10, 11, 16–34]; see Fig 1.

## Study characteristics

The 21 included meta-research studies were published from 2007 to 2021, representing 3,621 original studies or protocols and one survey with 106 participants; only three of these studies were published before 2013 [10, 18, 26]. The sample of the original study within each of the included meta-research studies varied. One meta-research study surveyed congress delegates [29], one study examined first-submission protocols for randomized controlled trials submitted to four hospital ethics committees [17], and 14 studies examined randomized or quasi-randomized primary studies published during a specific time period in a range of journals [10, 11, 18, 21–28, 31, 32, 34] or in specific databases [16, 19, 20, 30]. Finally, one study examined the

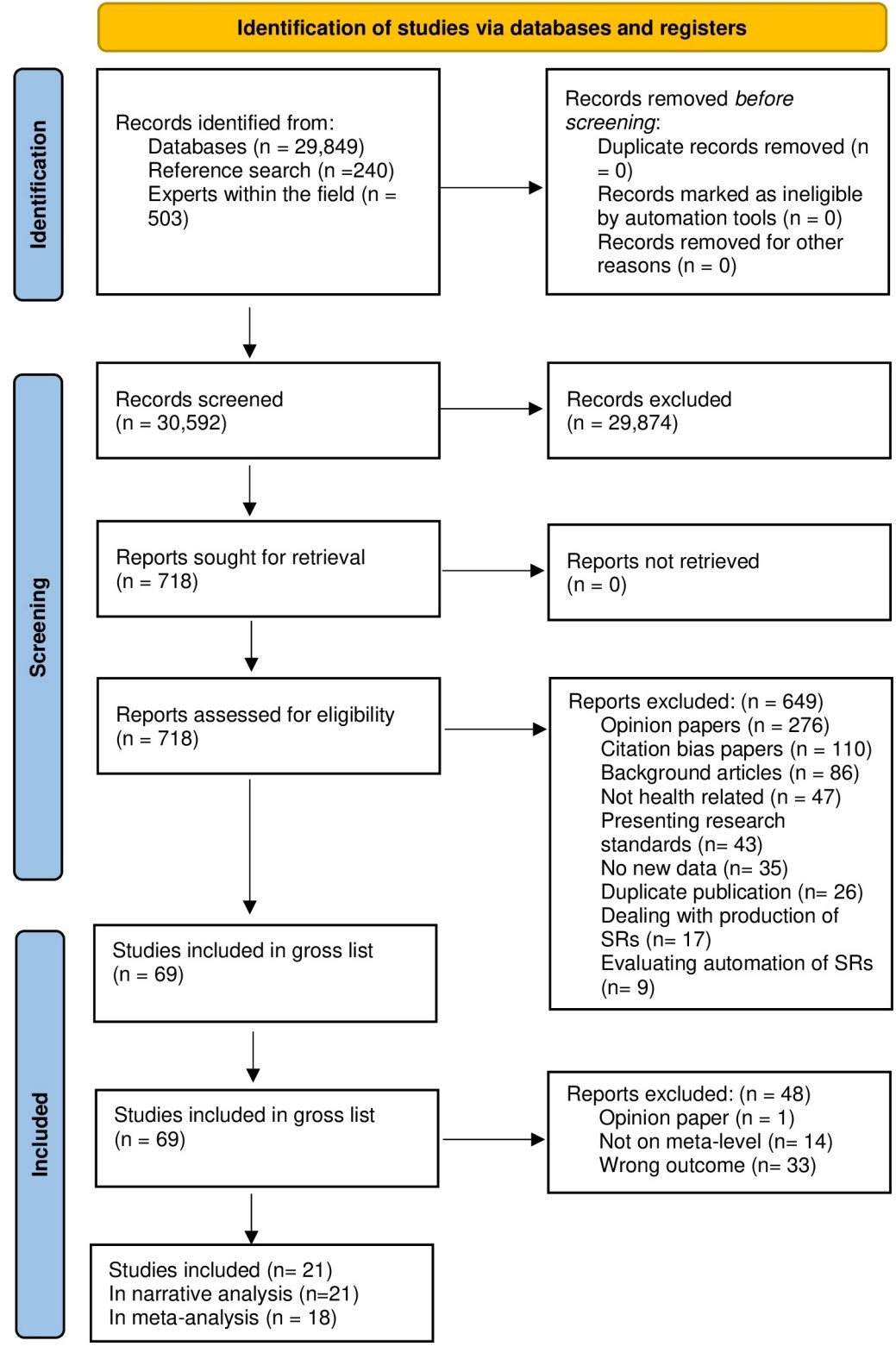

**Fig 1. PRISMA flow diagram.**

use of previously published systematic reviews when publishing a new systematic review [33]. Further, the number of original studies within each included meta-research study varied considerably, ranging from 18 [10] to 637 original studies [27]. The characteristics of the included meta-research studies are presented in Table 2.

## Risk of bias assessment

Overall, most studies were determined to exhibit a low risk of bias in the majority of items, and all of the included meta-research studies reported an unambiguous aim and a match between aim and methods. However, only a few studies provided argumentation for their choice of data source [17, 20, 24, 30], and only two of the 21 studies referred to an available a-priori protocol [16, 21]. Finally, seven studies provided poor or no discussion of the limitations of their study [10, 19, 22, 26–28, 34]. The risk-of-bias assessments are shown in Table 3.

## Synthesis of results

Of the included 21 studies, a total of 18 studies were included in the meta-analysis. Two studies included two cohorts each, and both cohorts in each of these studies were included in our meta-analysis [21, 30]. The survey by Clayton and colleagues, with a response rate of 17%, was not included in the meta-analysis as the survey did not provide data to identify the use of systematic reviews to justify specific studies. However, their results showed that 42 of 84 respondents (50%) reported using a systematic review for justification [29]. The study by Chow, which was also not included in the meta-analysis, showed that justification varied largely within and between specialties. However, only relative numbers were provided, and, therefore, no overall percentage could be extracted [11]. The study by Seehra et al. counted the SR citations in RCTs and not the number of RCTs citing SRs and is therefore not included in the meta-analysis either [23].

The percentage of original studies that justified a new study with a systematic review within each meta-research study ranged from 16% to 87%. The pooled percentage of original studies using systematic reviews to justify their research question was 42% (95% CI: 36% to 48%) as shown in Fig 2. Where the confidence interval showed the precision of the pooled estimate in a meta-analysis, the prediction interval showed the distribution of the individual studies. The heterogeneity in the meta-analysis assessed by $I^2$ was 94%. The clinical interpretation of this large heterogeneity is seen in a the very broad prediction interval ranging from 16 to 71%, meaning that based on these studies there is 95% chance that the results of the next study will show a prevalence between 16 to 71%.

Further, we conducted an explorative subgroup analysis of the study of Helfer et al. and the study of Joseph et al. as these two studies were on meta-analyses and protocols and therefore differ from the other included studies. This analysis did only marginally change the pooled percentage to 39% (95% CI; 33% to 46%) and the between-study variance ($tau^2$) was reduced with 23%.

The 21 included studies varied greatly in their approach and in their description of how systematic reviews were used, i.e., if the original studies referred and whether the used systematic reviews in the original studies were relevant and/or of high-quality. Nine studies assessed, to varying degrees, whether the used systematic reviews were relevant for the justification of the research [16–20, 25, 30, 32, 34]. Overall, the information reported by the meta-research studies was not sufficient to report the percentage of primary studies referring to relevant systematic reviews. No details were provided regarding the methodological quality of the systematic reviews used to justify the research question or if they were recently published reviews, except for Hoderlein et al., who reported that the mean number of years from publication of the cited systematic review and the trial report was four years [30].

**Table 2. Characteristics of the included meta–research studies (N = 21).**

| | Study aim | Study design | Material | Country | Inclusion period | Area of interest | Results | Conclusion |
|---|---|---|---|---|---|---|---|---|
| Bolland et al. (2018) | To investigate waste attributable to RCTs of supplementation in populations that were not vitamin D deficient and to determine the citation of SRs in large RCTs and protocols | Cross-sectional study of RCTs and protocols of RCTs | RCTs published in PubMed, ClinicalTrials.gov, the International Standard Randomised Controlled Trial Number (ISRCTN), the Australian New Zealand Clinical Trials Registry. Status survey data from published systematic reviews was supplemented by Medline, Embase, and Google searches | New Zealand | December 2015 | Vitamin D supplementation trials | When examining large RCTs and the citation of prior SRs of RCTs, three out of 18 studies referred to an SR to justify. Four out of the seven planned or ongoing trials with accessible relevant documents discuss SRs in their protocols or publications | Few large RCTs appeared to consider SRs in their design. Ongoing large RCTs share the same weaknesses of previous trials. Strategies to improve the design of RCTs should be introduced and studied to determine whether they can reduce research waste |
| Chapman et al. (2019) | The aim of the study was to quantify constituent components of waste in surgical RCTs and explore targets for improvement | Cross-sectional study of RCTs | RCTs registered in ClinicalTrials.gov and followed up by serial systematic searches of PubMed and Scopus databases were performed to determine publication status | UK | Between January 2011 and December 2012 | Surgery | Of 219 RCTs available for full-text review, 115 cited a relevant SR | A considerable burden of research waste in surgical RCTs. was identified. Future initiatives should target improvements in single-centre, poorly supported RCTs |
| Chow et al. (2017) | To quantify and summarize what types of evidence are cited in the introduction section as the reason for the RCT to be performed. The outcome was how many SRs were referred to when justifying the study | Cross-sectional analysis of RCTs. | Randomly chosen RCTs within six medical specialties | Canada | January-July 2015 | Medical fields: Ophthalmology, Otorthinola-ryngology, General surgery, Psychiatry, Obstetrics-gynaecology, Internal medicine | 148 RCTs were included (equally distributed between specialties). The different specialties cited SRs on average: Ophthalmology: 2.96, Otorhinolaryngology: 1.05, General Surgery: 1.40, Psychiatry: 1.16, Obs-Gyn: 0.68, and Internal Medicine: 1,11 SRs | Justifications for RCTs vary widely within and between specialties and the justification for conducting RCTs are not standardized |
| Clarke & Hopewell (2013) | To investigate whether SRs are used in the Introduction section | Cross-sectional analysis of RCTs | All RCTs published in the following 5 journals: Annals of Internal Medicine, BMJ, JAMA, The Lancet, NEJM | UK | May 2012 | No specific speciality | 35 RCTs were identified across the five journals and 13 studies (37%) referred to previous SRs in the introduction | Many trials still do not use SRs in their introduction |
| Clarke et al. (2007) | To assess to which extent reports began by referring to SRs to providing justification of the study | Cross-sectional analysis of RCTs | All RCTs published in the following 5 journals: Annals of Internal Medicine, BMJ, JAMA, The Lancet, NEJM | UK | May 2005 | No specific speciality | 18 RCTs were identified across the five journals and five studies referred to previous SRs in the introduction | Most researchers appear not to have considered an SR when justifying their trial |

*(Continued)*

**Table 2.** (Continued)

|  | Study aim | Study design | Material | Country | Inclusion period | Area of interest | Results | Conclusion |
|---|---|---|---|---|---|---|---|---|
| Clarke et al. (2010) | The which extent reports referred to SRs in their Introduction sections | Cross-sectional analysis of RCTs | All RCTs published in the following 5 journals: Annals of Internal Medicine, BMJ, JAMA, The Lancet, NEJM | UK | May 2009 | No specific specialty | 28 RCTs were identified across the five journals and one study used an updated SR and 10 studies referred to previous SRs in the introduction | Most researchers do not seem to have considered SRs when justifying a study. Findings have shown that editors and authors in five high-impact journals continue to fail to serve the needs of those who wish to use the results of RCTs to make decisions about health care |
| Clayton et al. (2017) | To summarize the current use of evidence synthesis in the trial design and analysis- The INVEST survey | Survey | Conference delegates at the 2-day International Clinical Trials Methodology Conference | UK | 16–17 November 2015 (own use the past 10 years) | No specific specialty | Of 638 registered, 106 completed the survey (17%). In total, 69 of 84 delegates had used a description of previous evidence to decide whether a trial is needed. 42 of 84 relevant respondents reported to have used a MA to justify a study | Trial teams responding to the INVEST survey generally reported that they are using evidence synthesis in trial design and analysis |
| De Meulemeester et al. (2018) | To assess whether recent RCTs meet scientific criteria, hypothesis use and SR use | Cross-sectional analysis of RCTs | All published RCTs in NEJM and JAMA in 2015 | Canada | 2015 | No specific specialty | 208 RCTs and 87 cited a relevant MA or SR in the published paper. | The majority of the published RCTs may not be scientifically and hence ethically justified |
| Engelking et al. (2018) | To analyse whether existing SRs were mentioned in RCTs published in journals as a rationale for conducting trial and for discussing results | Cross-sectional analysis of RCTs | RCTs published in in seven journals: Anaesthesia, Anaesthesia and Analgesia, An-anaesthesiology, Pain, British Journal of Anaesthesia, European Journal of Anaesthesiology, Regional Anaesthesia and Pain Medicine | Croatia | 2014–16 | Anaesthesiology | 622 RCTs included of which 278 cited an SR to justify the trial | Less than a fifth of trials published in high-impact journals in the field of anaesthesiology explicitly mention a previous SR as a justification for conducting the trial |
| Goudie et al. (2010) | To assess the extent to which authors currently make use of previous trial evidence in the design, analysis and reporting of RCTs | Cross-sectional analysis of RCTs | RCTs published in JAMA and Archives of Internal Medicine | UK | 5 months (January-May) 2007 | No specific specialty | 27 RCTs included and nine studies cited an SR | Consulting previous research before embarking a new trial and basing it on the impact of an up-dated MA will make reporting and designing more efficient |

(*Continued*)

**Table 2.** (*Continued*)

| | Study aim | Study design | Material | Country | Inclusion period | Area of interest | Results | Conclusion |
|---|---|---|---|---|---|---|---|---|
| Helfer et al. (2015) | To investigate whether MAs published in leading medical journals present an outline of available evidence by referring to previous MAs and SRs on the same topic | Cross-sectional citation analysis of SRs | SRs published in NEJM, The Lancet, JAMA, Annals of Internal Medicine, PLOS Medicine, British Medical Journal | Germany | Search completed in March 2013 | No specific specialty | 52 MAs were included and 45 cited a recent meta-analysis. | SRs on pharmacological treatments do not consistently refer to previous SRs. Can lead to research waste as only 2/3 of previous MA/SRs were cited |
| Hoderlein et al. (2017) | To investigate the extent to which RCTs of clinical trials of physiotherapy interventions use high-quality clinical research to justify the need for the trial | Cross-sectional analysis of RCTs | Random selected sample of clinical trials from Physiotherapy Evidence Database (PEDro) (10% of all studies in year 2001 and 2015) N = 70 and 151 | Germany | in 2001 and 2015 | Physiotherapy | N = 70 in 2001 and 151 studies in 2015 were included. 14 studies and 76 studies did cite an SR in 2001 and 2015, respectively | Many clinical trials of physiotherapy interventions are reported without reference to an SR in the introduction as justification for the study |
| Johnson et al. (2020) | To evaluate the use of SRs to justify RCTs and to analyze the reference of SRs for trial justification in RCTs | Cross-sectional analysis of RCTs | RCTs published in three high-ranking orthopaedic trauma journals, and for comparison RCTs published in general orthopaedic journals were used | US | January 1, 2015 to November 30, 2018 | Orthopaedia | 128 trauma RCTs included, and 30 studies cited an SR<br><br>Comparison:319 RCTs included and 115 cited an SR as justification for conducting the trial | Systematic reviews are frequently cited in orthopaedic trauma RCTs but are not commonly cited as justification for conducting a clinical trial |
| Joseph et al. (2018) | To evaluate the completeness of key domains in trial protocols involving children | Cross-sectional study of trial protocols | Four hospital Ethics committee Pharmacological trials proposals, involving children, submitted to the hospital ethics committees | Australia | Jan-December 2012 | Four hospital Ethics committee-clinical trials in children | 69 protocols included of which 33 referred to an SR | Protocols of clinical trials involving children omit many key domains |
| Ker & Roberts (2015) | To assess whether apparent redundancy in a cumulative meta-analysis is justified and to review trial justification | SRs with a cumulative meta-analysis including a qualitative review of trial justification of RCTs by SRs | Tranexamic acid (TXA) on surgical bleeding. MEDLINE, Embase, Cochrane central register, WHO int. trials registry platform up until May 2014 | UK | Updated the search to May 2014 | Tranexamic acid (TXA) on surgical bleeding | 118 studies included and 68 of these studies referred to an SR as reason for initiating at trial | Results indicate that poor quality is a more important cause of redundant research than the failure to review existing evidence |

(*Continued*)

**Table 2.** (*Continued*)

| | Study aim | Study design | Material | Country | Inclusion period | Area of interest | Results | Conclusion |
|---|---|---|---|---|---|---|---|---|
| Rauh et al. (2020) | To analyze published articles for citation of SRs for justification of conducting RCTs | Cross-sectional analysis of RCTs | RCTs published in PubMed in Obstetrics and Gynecology journals | US | January 1, 2014 – December 31, 2017 | Obstetrics and Gynecology | 458 included publications 279 (60.92%) cited an SR in the Introduction | A large portion of the RCTs recently published in clinical obstetrics and gynecology journals are not citing SRs as justification for conducting their studies, which may be leading to an increase in research waste |
| Rosenthal et al. (2017) | To investigate the use of SRs to inform trial design and for overall evidence synthesis | Cross-sectional analysis of RCTs | RCTs published in all issues of Annals of Surg., JAMA Surg. and British Journal of Surg. In 2010 | Switzer-land | 2010 | Surgical trials | 51 studies included– 8 studies referred to an SR in the Introduction | Results show that two thirds of the RCTs referenced an SR, however a few to justify or design or put results in context |
| Seehra et al. (2021) | To assess the extent to which reports of dental RCTs cite prior systematic reviews (SR) to explain the rationale or justification of the trial | Cross-sectional study of RCTs | An electronic database search in MEDLINE was undertaken to identify dental RCTs | UK | Between 1st January 2014 and 31st December 2019 | Dental Specialty Journals | 682 RCTs were included and 321 SRs were available for citation of which 62.5% did cite one of the SRs in the introduction | A relatively high proportion of dental RCTs (37.5%) did not cite an SR in the introduction section to justify the rationale of the trial when a relevant SR was available |
| Shepard et al. (2021) | To appraise the use of SRs as justification in RCTs and to report the manner in which SRs were incorporated into RCT manuscripts | Cross-sectional of RCTs | RCTs published in the top four urology journals based on Google Scholar h5 index | US | November 30,2014 – November 30 2019 | Urology | Of the 276 included RCTs, 169 cited an SR | RCTs published in four high impact urology journals inconsistently referenced an SR as justification |
| Torgerson et al. (2020) | To evaluate the use of SRs to justify conducting a RCT in top ophthalmology and optometry journals | Cross-sectional of RCTs | RCTs published in the top five Google Scholar h-5 index journals in Ophthalmology and Optometry | US | December 5, 2018 | Ophthalmology and Optometry | 152 RCTs included of which 41 cited an SR | Placing a higher priority on justifying RCTs with SRs would minimize research waste within ophthalmology |
| Walters et al. (2019) | To evaluate whether RCTs referenced SRs as the justification for conducting a trial | Cross-sectional study of RCTs | RCTs published in three high impact factor general medicine journals (NEJM, Lancet, JAMA) | US | January 1, 2016 – August 31, 2018 | General medicine | 637 RCTs were included and 243 cited an SR for trial justification | Less than half of the trials cited an SR as the basis for undertaking the trial |

SR: systematic review; MA: meta–analysis; RCT: randomized controlled trial.

**Table 3. Risk of bias of the included meta–research studies N = 21.**

| Study | 1. Clear and focused aim | 2. Match between aim and method(s) | 3 The best data source(s) chosen | 4. All important variables considered | 5. The same variables considered in all data sources | 6. Data collection transparent and data unambiguously identified | 7. Classification of the variables unaffected of prior knowledge about the results | 8. Appropriate analysis method | 9. Systematic error(s) or bias taken into consideration | 10. Conclusion supported by data |
|---|---|---|---|---|---|---|---|---|---|---|
| Bolland et al. (2018) | Low risk | Low risk | Unclear | Unclear | Low risk | Low risk | High risk | Low risk | High risk | Low risk |
| Chapman et al. (2019) | Low risk | Low risk | Low risk | Low risk | Low risk | Low risk | High risk | Low risk | Low risk | Low risk |
| Chow et al. (2017) | Low risk | Low risk | Unclear risk | Low risk | Low risk | Low risk | High risk | High risk | Low risk | Low risk |
| Clarke & Hopewell (2013) | Low risk | Low risk | Unclear risk | Low risk | Low risk | Low risk | High risk | Low risk | High risk | Low risk |
| Clarke et al. (2007) | Low risk | Low risk | Unclear risk | Unclear risk | Low risk | Low risk | High risk | Low risk | High risk | Low risk |
| Clarke et al. (2010) | Low risk | Low risk | Unclear risk | Low risk | Low risk | Low risk | High risk | Low risk | High risk | Low risk |
| Clayton et al. (2017) (survey) | Low risk | Low risk | High risk | Low risk | Low risk | Not applicable | High risk | Low risk | Low risk | High risk |
| De Meulemeester et al. (2018) | Low risk | Low risk | Unclear risk | Low risk | Low risk | Low risk | High risk | Low risk | Low risk | Low risk |
| Engelking et al. (2018) | Low risk | Low risk | Unclear risk | Low risk | Low risk | Unclear | High risk | Low risk | Low risk | Low risk |
| Goudie et al. (2010) | Low risk | Low risk | Unclear risk | Low risk | Low risk | Unclear | High risk | Low risk | Low risk | Low risk |
| Helfer et al. (2015) | Low risk | Low risk | Unclear risk | Low risk | Low risk | Low risk | High risk | Low risk | Low risk | Low risk |
| Hoderlein et al. (2017) | Low risk | Low risk | Low risk | Low risk | Low risk | Low risk | High risk | Low risk | Low risk | Low risk |
| Johnson et al. (2020) | Low risk | Low risk | Unclear risk | Low risk | Low risk | Low risk | Low risk | Low risk | Low risk | Low risk |
| Joseph et al. (2018) | Low risk | Low risk | Low risk | Low risk | Low risk | Unclear | High risk | Low risk | Low risk | Low risk |
| Ker K & Roberts I. (2015) | Low risk | Low risk | Unclear risk | Low risk | Low risk | High risk | Low risk | Low risk | Low risk | Low risk |
| Rauh et al. (2020) | Low risk | Low risk | Unclear risk | Unclear risk | Low risk | Low risk | High risk | Low risk | High risk | Low risk |
| Rosenthal et al. (2017) | Low risk | Low risk | Unclear risk | Unclear | Low risk | Low risk | High risk | Low risk | High risk | Low risk |
| Seehra et al. (2021) | Low risk | Low risk | Unclear risk | Low risk | Low risk | Low risk | High risk | Low risk | Low risk | Low risk |
| Shepard et al. (2021) | Low risk | Low risk | Low risk | Low risk | Low risk | Low risk | High risk | Low risk | Low risk | Low risk |
| Torgeson et al. (2020) | Low risk | Low risk | Unclear risk | Low risk | Low risk | Low risk | High risk | Low risk | Low risk | Low risk |
| Walters et al. (2019) | Low risk | Low risk | Unclear risk | Low risk | Low risk | Low risk | High risk | Low risk | High risk | Low risk |

## Discussion

We identified 21 meta-research studies, spanning 15 publication years and 12 medical disciplines. The findings showed substantial variability in the use of systematic reviews when

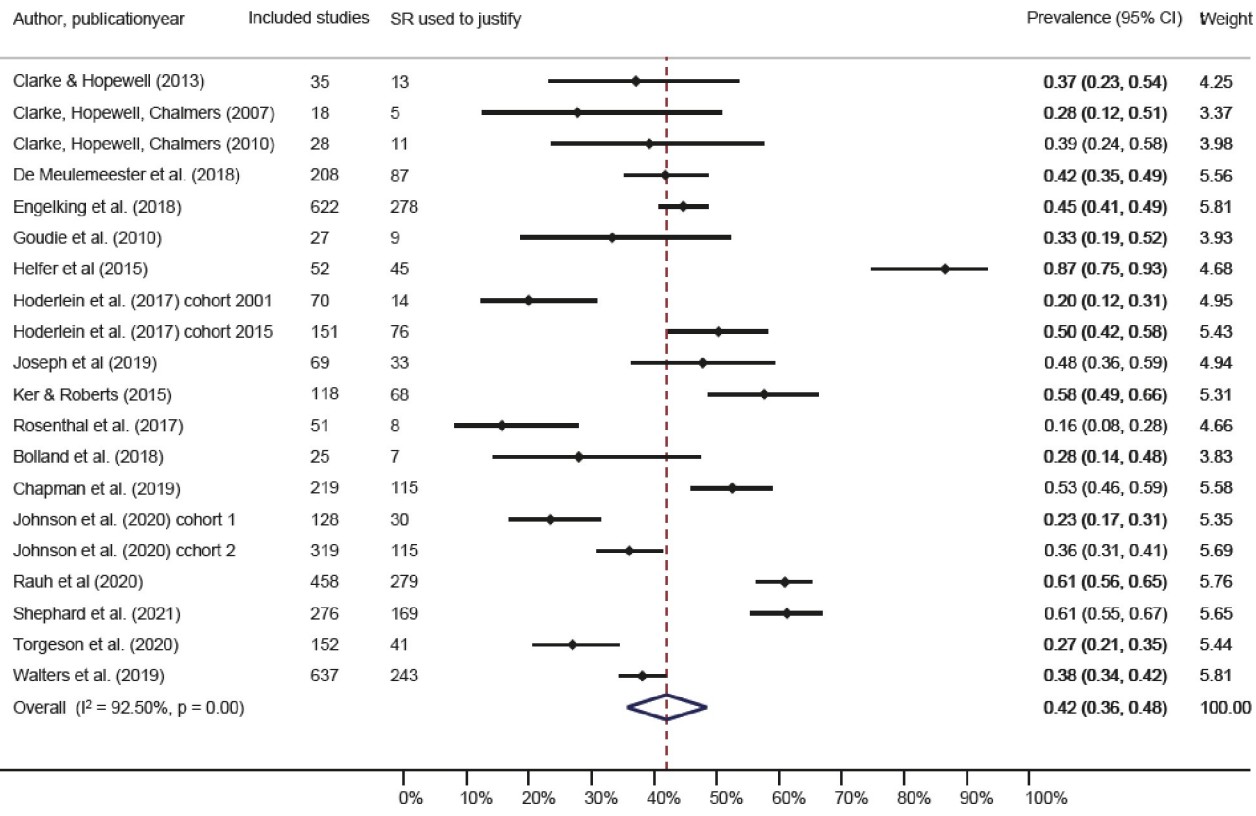

**Fig 2. Forest plot.** Forest plot prevalence and 95% confidence intervals for the percentage of studies using an SR to justify the study.

justifying new clinical studies, with the incidence of use ranging from 16% to 87%. However, fewer than half of the 19 meta-analysis-eligible studies used a systematic review to justify their new study. There was wide variability, and a general lack of information, about how systematic reviews were used within many of the original studies. Our systematic review found that the proportion of original studies justifying their new research using evidence syntheses is suboptimal and, thus, the potential for research redundancy continues to be a challenge. This study corroborates the serious possible consequences regarding research redundancy previously problematized by Chalmers et al. and Glasziou et al. [35, 36].

Systematic reviews are considered crucial when justifying a new study, as is emphasized in reporting guidelines such as the CONSORT statement [37]. However, there are challenges involved in implementing an evidence-based research approach. The authors of the included meta-research study reporting the highest use of systematic reviews to justify a new systematic review study point out that even though the authors of the original studies refer to some of the published systematic reviews, they neglect others on the same topic, which may be problematic and result in a biased approach [33]. Other issues that have been identified are the risk of research waste when a systematic review may not be methodologically sound [12, 38] and that there is also redundancy in the conduct of systematic reviews, with many overlapping systematic reviews existing on the same topic [39–41]. In the original studies within the meta-research studies, the use of systematic reviews was not consistent and, further, it was not explicated whether the systematic reviews used were the most recent and/or of high methodological quality. These issues speak to the need for refinement in the area of systematic review development, such as mandatory registration in prospective registries. Only two out of the included 21

studies in this study referred to an available a-priori protocol [16, 21]. General recommendations in the use of systematic reviews as justification for a new study are difficult as these will be topic specific, however researchers should be aware to use the most robust and methodologically sound of recently published reviews, preferably with á priori published protocols.

Efforts must continue in promoting the use of evidence-based research approaches among clinical health researchers and other important stakeholders, such as funders. Collaborations such as the Ensuring Value in Research Funders Forum, and changes in funding review criteria mandating reference to previously published systematic reviews when justifying the research question within funding proposals, are examples of how stakeholders can promote research that is evidence-based [8, 41].

## Strengths and limitations

We conducted a comprehensive and systematic search. The lack of standard terminology for meta-research studies resulted in search strategies that retrieved thousands of citations. We also relied on snowballing efforts to identify relevant studies, such as by contacting experts and scanning the reference lists of relevant studies.

There is also a lack of tools to assess risk of bias for meta-research studies, so a specific risk-of bias tool for the five conducted reviews was created. The tool was discussed and revised continuously throughout the research process; however, we acknowledge that the checklist is not yet optimal and a validated risk-of-bias tool for meta-research studies is needed.

Many of the included meta-research studies did not provide details as to whether the systematic reviews used to justify the included studies were relevant, high-quality and/or recently published. This may raise questions as to the validity of our findings, as the majority of the meta-research studies only provide an indication of the citation of systematic reviews to justify new studies, not whether the systematic review cited was relevant, recent and of high-quality, or even how the systematic review was used. We did not assess this further either. Nonetheless, even if we assumed that these elements were provided for every original study included in the included meta-research studies (i.e. taking a conservative approach), fewer than half used systematic reviews to justify their research questions. The conservative approach used in this study therefore does not underestimate, and perhaps rather overestimates, the actual use of relevant systematic reviews to justify studies in clinical health science across disciplines.

Different study designs were included in the meta-analysis, which may have contributed to the high degree of heterogeneity observed. Therefore, the presented results should be interpreted with caution due to the high heterogeneity. Not only were there differences in the methods of the included meta-research studies, but there was also heterogeneity in the medical specialties evaluated [42, 43].

## Conclusion

In conclusion, justification of research questions in clinical health research with systematic reviews continues to be inconsistent; fewer than half of the primary studies within the included meta-research studies in this systematic review were found to have used a systematic review to justify their research question. This indicates that the risk of redundant research is still high when new studies across disciplines and professions in clinical health are initiated, thereby indicating that evidence-based research has not yet been successfully implemented in the clinical health sciences. Efforts to raise awareness and to ensure an evidence-based research approach continue to be necessary, and such efforts should involve clinical health researchers themselves as well as important stakeholders such as funders.

## Supporting information

**S1 Checklist.**
(DOCX)

**S1 Protocol.**
(PDF)

**S1 File.**
(DOCX)

**S1 Data.**
(XLSX)

## Acknowledgments

This work has been prepared as part of the Evidence-Based Research Network (ebrnetwork. org). The Evidence-Based Research Network is an international network that promotes the use of systematic reviews when justifying, designing, and interpreting research. The authors thank the Section for Evidence-Based Practice, Department for Health and Function, Western Norway University of Applied Sciences for their generous support of the EBRNetwork. Further, thanks to COST Association for supporting the COST Action "EVBRES" (CA 17117, evbres. eu) and thereby the preparation of this study. Thanks to Gunhild Austrheim, Head of Unit, Library at Western Norway University of Applied Sciences, Norway, for helping with the second search. Thanks to those helping with the screening: Durita Gunnarsson, Gorm Høj Jensen, Line Sjodsholm, Signe Versterre, Linda Baumbach, Karina Johansen, Rune Martens Andersen, and Thomas Aagaard.

We gratefully acknowledge the contribution from the EVBRES (COST ACTION CA 17117) Core Group, including Anne Gjerland (AG) and her specific contribution to the search and screening process.

## Author Contributions

**Conceptualization:** Jane Andreasen, Birgitte Nørgaard, Eva Draborg, Carsten Bogh Juhl, Jennifer Yost, Klara Brunnhuber, Karen A. Robinson, Hans Lund.

**Data curation:** Jane Andreasen, Birgitte Nørgaard, Eva Draborg, Carsten Bogh Juhl, Hans Lund.

**Formal analysis:** Carsten Bogh Juhl.

**Methodology:** Jane Andreasen, Birgitte Nørgaard, Eva Draborg, Carsten Bogh Juhl, Jennifer Yost, Klara Brunnhuber, Karen A. Robinson, Hans Lund.

**Project administration:** Jane Andreasen, Hans Lund.

**Supervision:** Hans Lund.

**Writing – original draft:** Jane Andreasen.

**Writing – review & editing:** Jane Andreasen, Birgitte Nørgaard, Eva Draborg, Carsten Bogh Juhl, Jennifer Yost, Klara Brunnhuber, Karen A. Robinson, Hans Lund.

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
