## [Decision Letter · Decision Letter 0]

2 Apr 2022

PONE-D-22-02383Justification of research using systematic reviews continues to be inconsistent in clinical health science - a systematic review and meta-analysis of meta-research studiesPLOS ONE

Dear Dr. Andreasen,

Thank you for submitting your manuscript to PLOS ONE. After careful consideration, we feel that it has merit but does not fully meet PLOS ONE’s publication criteria as it currently stands. Therefore, we invite you to submit a revised version of the manuscript that addresses the points raised during the review process.

We look forward to receiving your revised manuscript.

Kind regards,

Andrzej Grzybowski

Academic Editor

PLOS ONE

Journal Requirements:

- https://www.jclinepi.com/article/S0895-4356(22)00016-6/fulltext

In your revision ensure you cite all your sources (including your own works), and quote or rephrase any duplicated text outside the methods section. Further consideration is dependent on these concerns being addressed.

Reviewers' comments:

Reviewer's Responses to Questions

**Comments to the Author**

1. Is the manuscript technically sound, and do the data support the conclusions?

Reviewer #1: Yes

Reviewer #2: Yes

2. Has the statistical analysis been performed appropriately and rigorously? 

Reviewer #1: Yes

Reviewer #2: Yes

3. Have the authors made all data underlying the findings in their manuscript fully available?

Reviewer #1: Yes

Reviewer #2: Yes

4. Is the manuscript presented in an intelligible fashion and written in standard English?

Reviewer #1: Yes

Reviewer #2: Yes

5. Review Comments to the Author

Reviewer #1: Thank you for the opportunity to review this interesting meta-research paper, which is part of a series of papers.

Basing new research on systematic reviews is clearly important and has been the subject of a number of reviews. This paper essentially reviews the meta-research in this area, to give a global assessment of the issue taking into account all of the evidence

The content of the rest of the series was not made clear, but a decision has been made to publish them singly. I think the short description of the rest of the programme could be expanded a little to put the work in context and help the reader understand how the work fits together. How do the different studies relate, and are other papers needed to put the current work in context?

The introduction defines meta-research in broad terms, but it is not until the results that the reader is given a sense of the actual designs included and of relevance to the research question. Were these defined a priori, or were these study designs that fit the broad definition which happened to be found in the search? Are there meta-research designs of relevance to the research question which were not found in the searches?

Personally, I would bring a description of the range of study design forward into the introduction, as getting a sense of the sorts of approaches to meta-research of relevance will help non-specialists in this area. I was not clear of the likely designs until quite late in the paper

The review methods seemed very rigorous, and I had no major comments on those beyond one clarification. When they said, ‘No study was excluded on the grounds of low quality’, did they mean that no studies were considered so bad, or that as a rule no studies were every going to be excluded on that basis?

As noted above, there were a number of study designs included, and all were assessed using the generic risk of bias tool. Presumably some designs are just stronger than others? The survey must be considered a weaker design that the others. Again, this links to the earlier comment about the need for more detail on design of the meta research, which I felt was lost in the use of a generic risk of bias assessment.

I did not understand the statement ‘The clinical interpretation of the large heterogeneity is seen in a broad prediction interval with a range from 16 to 71%’ and that needs clarification

The discussion is balanced, but there are a few significant issues that are given a fairly cursory consideration and would benefit from greater detail

I was interested in the issue of the ‘quality’ of the reviews used. I accept that the data here was not enough for analysis, but felt that the authors (as experts in this area) could be pushed to provide a stronger statement about what criteria should be used by further studies (for example, how do we judge if a review used as the basis for research is a strong basis. How long before a quoted review is too ‘old’?)

They acknowledge that ‘the checklist is not yet optimal and a validated risk-of-bias tool for meta-research studies is needed’. Given their experience and expertise, what would that look like, and how would it be best developed and tested? How would it take into account the role of different designs noted above, given variation in the approaches to meta-research they found?

I appreciate the simple and elegant assessment of the main findings, but they present only vague statement on the role of design and medical specialities. Is it not possible for them to say more on this, or explore the data more fully? What about change over time, which seems very relevant. I did feel the authors could be pushed a little more here, given that they have a programme of work and must be in a position to present more substantive statements. I think that would add to the contribution of the paper

Reviewer #2: The article is on interesting topic but several points needs emphasis:

the inclusion criteria should be defined more clearly in the text

Systematiic review and meta analysis are relatively new and first papers go to late seventies in previous century.

This should be considered when reviewing papers.

The risk of redundancy could not be well defined from the meta search papers rather it should be from the original articles . This would not be possible unless a focused issue is chosen as an example.

The different disciplines have different research out puts as the basis for systematic reviews which makes the comparison difficult .

I realize some studies are based on the disclosure of the authors whether they have used the previous systematic reviews or not . This should be confirmed by evidence .

These should be mentioned as the limitations of this work .

6. PLOS authors have the option to publish the peer review history of their article (what does this mean?). If published, this will include your full peer review and any attached files.

Reviewer #1: **Yes: **Peter Bower

Reviewer #2: No

---

## [Author Response · Author response to Decision Letter 0]

25 Apr 2022

Response letter to the editor and reviewers,

Thank you for the opportunity to revise the manuscript. Thank you to the reviewers for the positive and constructive comments concerning the manuscript. We have now revised the manuscript in accordance with these comments by addressing all issues from the editor and from the reviewers below.

Journal Requirements:

Answer: we have addressed the requirements, see our answers below.

Answer: We believe we meet the style requirements, including correct file naming.

- https://www.jclinepi.com/article/S0895-4356(22)00016-6/fulltext

In your revision ensure you cite all your sources (including your own works), and quote or rephrase any duplicated text outside the methods section. Further consideration is dependent on these concerns being addressed.

Answer: We agree that there are overlap in parts of the methods section with the mentioned publication. The paper was published in the period of this manuscript being in review, we have therefore now referred to the publication in this manuscript. This manuscript and the publication are both part of a series of papers assessing the global status of evidence-based research in clinical health research and therefore the overlap in the methods section was expected. We have thoroughly scrutinized the full manuscript and found no full sentences that are overlapping, except for the methods section. To be sure of this, we further have conducted a legal comparison in MS Words with the mentioned publication and again found no full sentences except in the methods section. This is to our sincere knowledge only in the methods section, please let us know if we are mistaken.

Answer: We have uploaded the data set necessary to replicate our study findings in a supplementary file and described the changes to the “Data Availability statement” in the cover letter.

Answer: We have uploaded the data set necessary to replicate our study findings in a supplementary file and described the changes to the “Data Availability statement” in the cover letter.

5. Review Comments to the Author

Reviewer comments Reviewer #1: 

1. Thank you for the opportunity to review this interesting meta-research paper, which is part of a series of papers.

Basing new research on systematic reviews is clearly important and has been the subject of a number of reviews. This paper essentially reviews the meta-research in this area, to give a global assessment of the issue taking into account all of the evidence

Response: Thank you for this response and that is exactly the purpose.

2. The content of the rest of the series was not made clear, but a decision has been made to publish them singly. I think the short description of the rest of the program could be expanded a little to put the work in context and help the reader understand how the work fits together. How do the different studies relate, and are other papers needed to put the current work in context?

Response: We have expanded the text and especially regarding how the work fits together and shows our purpose of taking a global assessment of the on evidence-based research in the following six papers:

1. Meta-research evaluating redundancy and use of systematic reviews when planning new studies in health research – a scoping review

2. A Systematic Review on the Use of Prior Research in Reports of Randomized Clinical Trials

3. Justification

4. Design

5. Context

6. The problem of citation bias – a scoping review

We do not have other papers in pipeline at the moment, but we are currently working on a Handbook for Evidence-Based Research to provide tools and models to make it easier for researchers to work evidence- based in their research.

Changes to text: This study is one of six ongoing meta-syntheses (four systematic reviews and two scoping reviews) planned to assess the global state of evidence-based research in clinical health research. These are; a scoping review mapping the area broadly to describe current practice and identify knowledge gaps, a systematic review on the use of prior research in reports of randomized controlled trials specifically, three systematic reviews assessing the use of systematic reviews when justifying, designing [14] or putting results of a new study in context, and finally a scoping review uncovering the breadth and characteristics of the available, empirical evidence on the topic of citation bias . Further, the research group is working with colleagues on a Handbook for Evidence-based Research in health sciences.

3. The introduction defines meta-research in broad terms, but it is not until the results that the reader is given a sense of the actual designs included and of relevance to the research question. Were these defined a priori, or were these study designs that fit the broad definition which happened to be found in the search? Are there meta-research designs of relevance to the research question which were not found in the searches?

Response: We get your point. A very broad and inclusive definition was defined a priori in the published protocol: “Types of study to be included: We will include meta-research studies (or studies performing research on research)” in order not to miss out on relevant studies, because the research field was quite new and further, we did not identify other meta-research studies to guide our process. Due to our very broad and sensitive search strategy we believe we identified all relevant meta-research studies. 

Only data regarding justification from original papers were included in our meta-analysis as the study design of a survey of delegates use of systematic reviews to justify their studies, was assessed as seriously subjected to a social desirability bias.

Changes to text: 

Introduction: The present systematic review aimed to identify and synthesize results from meta-research studies, regardless study type, evaluating if and how authors of clinical health research studies use systematic reviews to justify a new study.

Methods section, eligibility criteria: Studies were eligible for inclusion if they were original meta-research studies, regardless study type, that evaluated if and how authors of clinical health studies used systematic reviews to justify new clinical health studies.

4. Personally, I would bring a description of the range of study design forward into the introduction, as getting a sense of the sorts of approaches to meta-research of relevance will help non-specialists in this area. I was not clear of the likely designs until quite late in the paper

Response: We agree and have made it clear that all meta-research studies regardless design was included. 

Changes to text: see above.

5. The review methods seemed very rigorous, and I had no major comments on those beyond one clarification. When they said, ‘No study was excluded on the grounds of low quality’, did they mean that no studies were considered so bad, or that as a rule no studies were every going to be excluded on that basis?

Response: The latter, as a rule no studies were excluded, as our intention was not to guide clinical practice. This is stated in the manuscript as the last sentence in the Risk-of-Bias Assessment section. No changes are therefore made.

6. As noted above, there were a number of study designs included, and all were assessed using the generic risk of bias tool. Presumably some designs are just stronger than others? The survey must be considered a weaker design that the others. Again, this links to the earlier comment about the need for more detail on design of the meta research, which I felt was lost in the use of a generic risk of bias assessment.

Response: We agree on this point, but we did take a very open approach to monitor the field of justification. And we did not range the study designs in a hierarchical order in our “premature” Risk of Bias tool, as we aimed to assess the area and not to provide any clinical recommendations. However, the author group and colleagues are currently working on an improved checklist tool.

No further changes to text.

7. I did not understand the statement ‘The clinical interpretation of the large heterogeneity is seen in a broad prediction interval with a range from 16 to 71%’ and that needs clarification

Response: We agree that an explanation is appropriate.

Changes to text: The clinical interpretation of the large heterogeneity is seen in a broad prediction interval with a range from 16 to 71%, meaning that there is 95% confidence that the results of the next study will be between a prevalence of 16 to 71%. 

8. The discussion is balanced, but there are a few significant issues that are given a fairly cursory consideration and would benefit from greater detail

Response: We have addressed the issues mentioned below and provided more detail

9. I was interested in the issue of the ‘quality’ of the reviews used. I accept that the data here was not enough for analysis, but felt that the authors (as experts in this area) could be pushed to provide a stronger statement about what criteria should be used by further studies (for example, how do we judge if a review used as the basis for research is a strong basis. How long before a quoted review is too ‘old’?)

Response: Very interesting topic to address further, which we have continuously discussed in the author group, but this is both complex and context dependent in specific topics. Therefore, we have chosen not to elaborate further on the topic in the manuscript, to give an appropriate consideration more space is needed.

Instead, we have mentioned these considerations as important to address further in future publications as to guide researchers when using systematic reviews to justify. As mentioned earlier, the research group is working with colleagues on a Handbook for Evidence-based Research in health sciences, which will elaborate on the topics in detail.

Changes to text in Discussion section:

General recommendations in the use of systematic reviews as justification for a new study are difficult as these will be topic specific, however researchers should be aware to use the most robust and methodologically sound of recently published reviews, preferably with á priori published protocols.

10. They acknowledge that ‘the checklist is not yet optimal and a validated risk-of-bias tool for meta-research studies is needed’. Given their experience and expertise, what would that look like, and how would it be best developed and tested? How would it take into account the role of different designs noted above, given variation in the approaches to meta-research they found?

Response: We fully agree with you on this topic and the author group and colleagues are currently working on an improved checklist tool. Your suggestion about ranging the study designs is very relevant and will be considered in the author group in this thorough work that we expect to publish in the near future. We find the work requires space and thorough analysis and we therefore have decided this should be published in an independent paper. 

11. I appreciate the simple and elegant assessment of the main findings, but they present only vague statement on the role of design and medical specialities. Is it not possible for them to say more on this, or explore the data more fully? What about change over time, which seems very relevant. I did feel the authors could be pushed a little more here, given that they have a programme of work and must be in a position to present more substantive statements. I think that would add to the contribution of the paper

Response: The role of design is only considered in relation to that the studies has done meta - research on the topic “justification”. We do not find it was appropriate to explicate more about the roles of medical specialties as the approach in the different studies were very diverse ranging from participants in the survey, to specialties or to specific journals (mostly high ranking) or more broad aimed journals or databases.

Change over time is an important and relevant question. We did not address the issue for two reasons. Firstly, most of the papers are published after 2012 and it would be a short timeline to assess. But most importantly, as most of the included studies in our meta-research study were cross-sectional, we would not be able to validly assess change over time with the data at hand.

Reviewer comments Reviewer #2 :

1. The article is on interesting topic but several points needs emphasis

Response: Thank you. We have answered each point above.

2. The inclusion criteria should be defined more clearly in the text

Response: Methods section: we have clarified the inclusion criteria in the methods section.

Changes to text: 

Methods section, eligibility criteria: Studies were eligible for inclusion if they were original meta-research studies, regardless study type, that evaluated if and how authors of clinical health studies used systematic reviews to justify new clinical health studies.

3. Systematic review and meta analysis are relatively new and first papers go to late seventies in previous century. This should be considered when reviewing papers.

Response: Yes, it is a fairly new discipline, however it has been recommended to be evidence-based by the use of systematic reviews and meta-analyses for many years. Our aim was therefore to look at meta-research in a broad sense by using previously published studies investigating how large a percentage are using systematic reviews as justification when initiating new health science. 

4. The risk of redundancy could not be well defined from the meta search papers rather it should be from the original articles . This would not be possible unless a focused issue is chosen as an example.

Response: Risk of redundancy can, in our perspective, be thoroughly assessed by the use of systematic reviews with meta-analyses included, and especially cumulative meta-analyses can pinpoint this in a specific research topic. Therefore, we agree that we cannot point it to a specific field but have taken this meta-research perspective to provide a more global status on the topic.

We hope you can follow our reasoning.

5. The different disciplines have different research out puts as the basis for systematic reviews which makes the comparison difficult

Response: In this paper, we did not look for the output, but the “input” so to speak, as we assess whether the authors have used justification by using systematic reviews, when initiating a new study in health science. We agree, it is important to define the aim and approach and the outcomes more specifically, if you look into a specific topic.

No changes to text.

6. I realize some studies are based on the disclosure of the authors whether they have used the previous systematic reviews or not. This should be confirmed by evidence.

These should be mentioned as the limitations of this work.

Response: We agree on this point and have clarified in the limitations that we have taken “the face value” reported by the authors in the included studies.

Changes to text: Discussion, Strengths and Limitations section:

This may raise questions as to the validity of our findings, as the majority of the meta-research studies only provide an indication of the citation of systematic reviews to justify new studies, not whether the systematic review was relevant, recent or of high-quality, or even how the systematic review was used. We did not assess this further either.

---

## [Decision Letter · Decision Letter 1]

19 Sep 2022

PONE-D-22-02383R1Justification of research using systematic reviews continues to be inconsistent in clinical health science - a systematic review and meta-analysis of meta-research studiesPLOS ONE

Dear Dr. Andreasen,

Thank you for submitting your manuscript to PLOS ONE. After careful consideration, we feel that it has merit but does not fully meet PLOS ONE’s publication criteria as it currently stands. Therefore, we invite you to submit a revised version of the manuscript that addresses the points raised during the review process.

We look forward to receiving your revised manuscript.

Kind regards,

Andrzej Grzybowski

Academic Editor

PLOS ONE

Journal Requirements:

Reviewers' comments:

Reviewer's Responses to Questions

**Comments to the Author**

1. If the authors have adequately addressed your comments raised in a previous round of review and you feel that this manuscript is now acceptable for publication, you may indicate that here to bypass the “Comments to the Author” section, enter your conflict of interest statement in the “Confidential to Editor” section, and submit your "Accept" recommendation.

Reviewer #1: All comments have been addressed

2. Is the manuscript technically sound, and do the data support the conclusions?

Reviewer #1: Yes

3. Has the statistical analysis been performed appropriately and rigorously? 

Reviewer #1: Yes

4. Have the authors made all data underlying the findings in their manuscript fully available?

Reviewer #1: Yes

5. Is the manuscript presented in an intelligible fashion and written in standard English?

Reviewer #1: Yes

6. Review Comments to the Author

Reviewer #1: I am happy with the responses and thank the authors for their detailed replies, but just had 2 minor issues

This probably reflects my ignorance so apologies to the authors, but I still do not understand the relationship between the 95% CI around the pooled percentage, and the 'broad prediction interval' which follows it. Could they add a line to explain?

There are some typos remaining. The phrase 'regardless study type' should read 'regardless of study type'. There are some rogue apostrophes in the tables (SR's, RCT's) which need to be edited

7. PLOS authors have the option to publish the peer review history of their article (what does this mean?). If published, this will include your full peer review and any attached files.

Reviewer #1: No

---

## [Author Response · Author response to Decision Letter 1]

21 Sep 2022

Response letter

Thank you for the opportunity to revise the manuscript. Thank you to the reviewer for the relevant comments concerning the manuscript. We have revised the manuscript in accordance with these comments by addressing all issues from the editor and from the reviewers below.

6. Review Comments to the Author

Reviewer #1: I am happy with the responses and thank the authors for their detailed replies, but just had 2 minor issues

Response: Thank you very much.

This probably reflects my ignorance so apologies to the authors, but I still do not understand the relationship between the 95% CI around the pooled percentage, and the 'broad prediction interval' which follows it. Could they add a line to explain?

Response: We have revised and explained more in detail and hope the revised text explains this more clearly.

Changes to text:

Where the confidence interval showed the precision of the pooled estimate in a meta-analysis, the prediction interval showed the distribution of the individual studies. The heterogeneity in the meta-analysis assessed by I2 was 94%. The clinical interpretation of this large heterogeneity is seen in a the very broad prediction interval ranging from 16 to 71%, meaning that based on these studies there is 95% chance that the results of the next study will show a prevalence between 16 to 71%. 

There are some typos remaining. The phrase 'regardless study type' should read 'regardless of study type'.

Response: Thank you, we have revised as suggested.

There are some rogue apostrophes in the tables (SR's, RCT's) which need to be edited

Response: Thank you for pointing this out. We have edited this now.

On behalf of the author group,

Jane Andreasen

---

## [Decision Letter · Decision Letter 2]

18 Oct 2022

Justification of research using systematic reviews continues to be inconsistent in clinical health science - a systematic review and meta-analysis of meta-research studies

PONE-D-22-02383R2

Dear Dr. Andreasen,

We’re pleased to inform you that your manuscript has been judged scientifically suitable for publication and will be formally accepted for publication once it meets all outstanding technical requirements.

Kind regards,

Andrzej Grzybowski

Academic Editor

PLOS ONE

Additional Editor Comments (optional):

Reviewers' comments:

Reviewer's Responses to Questions

**Comments to the Author**

1. If the authors have adequately addressed your comments raised in a previous round of review and you feel that this manuscript is now acceptable for publication, you may indicate that here to bypass the “Comments to the Author” section, enter your conflict of interest statement in the “Confidential to Editor” section, and submit your "Accept" recommendation.

Reviewer #1: All comments have been addressed

2. Is the manuscript technically sound, and do the data support the conclusions?

Reviewer #1: Yes

3. Has the statistical analysis been performed appropriately and rigorously? 

Reviewer #1: Yes

4. Have the authors made all data underlying the findings in their manuscript fully available?

Reviewer #1: Yes

5. Is the manuscript presented in an intelligible fashion and written in standard English?

Reviewer #1: Yes

6. Review Comments to the Author

Reviewer #1: (No Response)

7. PLOS authors have the option to publish the peer review history of their article (what does this mean?). If published, this will include your full peer review and any attached files.

Reviewer #1: **Yes: **Peter Bower

---

## [Editor Report · Acceptance letter]

21 Oct 2022

PONE-D-22-02383R2 

Justification of research using systematic reviews continues to be inconsistent in clinical health science - a systematic review and meta-analysis of meta-research studies 

Dear Dr. Andreasen:

I'm pleased to inform you that your manuscript has been deemed suitable for publication in PLOS ONE. Congratulations! Your manuscript is now with our production department. 

Kind regards, 

on behalf of

Dr. Andrzej Grzybowski 

Academic Editor

PLOS ONE